# ARE COMPLICATED LOSS FUNCTIONS NECESSARY FOR TEACHING LLMS TO REASON?

## ABSTRACT

Recent advances in large language models (LLMs) highlight the importance of post-training techniques for improving reasoning and mathematical ability. Group Relative Policy Optimization (GRPO) has shown promise in this domain by combining group-relative advantage estimation, PPO-style clipping, and KL regularization. However, its complexity raises the question of whether all components are necessary for fostering reasoning behaviors. We conduct a systematic analysis of GRPO and identify two key findings: (1) incorporating negative feedback is essential—training solely on actions above a baseline limits learning; and (2) PPO-style constraints, such as policy ratio clipping, are not required to improve mathematical reasoning or performance. Building on these insights, we propose REINFORCE with Group Relative Advantage (RGRA), a simplified variant that retains group-relative advantage estimation but removes PPO-style clipping and policy ratio terms. Experiments across standard mathematical benchmarks indicate that RGRA has the potential to achieve stronger performance than GRPO. Our results suggest that simpler REINFORCE-based approaches can effectively enhance reasoning in LLMs, offering a more transparent and efficient alternative to GRPO.

## 1 INTRODUCTION

Recent advancements in artificial intelligence have been largely driven by large language models (LLMs) (OpenAI et al., 2024; Team et al., 2025; Jiang et al., 2024), which demonstrate remarkable capabilities across a wide range of tasks, from text translation Becker et al. (2024) to complex mathematical problem-solving Wang et al. (2025). A key factor in their progress has been improvements in the post-training phase, which aligns model outputs more closely with human preferences and enhances task-specific performance.

To address this challenge, researchers introduced approaches such as Reinforcement Learning from Human Feedback (RLHF), which trains models using reward signals designed to approximate human prefernces. Early RLHF methods, like Proximal Policy Optimization (PPO) (Schulman et al., 2017b), relied on two separate models: a policy model (the LLM to be aligned), a reward model (Ouyang et al., 2022a), and a value model which adds computational and implementation complexity.

Recently, Group Relative Policy Optimization (GRPO) (Shao et al., 2024a) has emerged as a promising alternative. Introduced in Shao et al. (2024a), GRPO eliminates the need for a value model by estimating advantages as normalized rewards across a sampled group of completions for each prompt. This innovation simplifies the training pipeline while achieving strong performance on mathematical reasoning benchmarks. Building on this, the DEEPSEEK-R1 model (DeepSeek-AI et al., 2025) demonstrated that combining GRPO with simple rule-based rewards for format and correctness can elicit extended reasoning traces and complex reasoning behaviors.

Despite its empirical success, GRPO's loss function combines several components — group-relative advantage estimation, PPO-style clipping, and KL regularization — resulting in a level of complexity that may not be strictly necessary for effective learning. This observation is underscored by recent GRPO variants targeting scalability, stability, and efficiency: Prefix Grouper optimizes shared-prefix encoding for faster training (Liu et al., 2025b), CPPO prunes low-advantage completions to reduce sampling cost (Lin et al., 2025b), DAPO introduces an increased upper clipping range to increase

the exploration of the policy Yu et al. (2025), S-GRPO facilitates early exit, allowing the model to focus only on the essential reasoning steps (Dai et al., 2025), and GTPO addresses token-level conflicts and policy collapse by introducing trajectory-level protection mechanisms (Simoni et al., 2025). Together, these works highlight both the promise of GRPO-style optimization and the potential overengineering of its current formulation.

In this work, we take a complementary approach: rather than proposing yet another GRPO variant, we *systematically analyze and simplify* its loss function. By isolating and removing individual components, we aim to identify which elements are essential for enabling learning and which can be simplified or removed without substantially compromising performance. Furthermore, we evaluate whether simpler reinforcement learning methods — including variants of REINFORCE (Ahmadian et al., 2024) and Rejection Sampling Fine-Tuning (RAFT) (Liu et al., 2024) — can match or surpass the performance improvements achieved by GRPO-trained models on mathematical reasoning tasks. This analysis contributes both conceptual clarity and practical guidance for designing efficient and robust post-training strategies for reasoning-focused LLMs.

## 2 BACKGROUND

### 2.1 RELATED WORK

**Reasoning in Large Language Models.** Recent advancements in large language models (LLMs) have shown that, once models reach sufficient scale, they exhibit emergent behaviors, including the capacity for reasoning (Wei et al., 2022). Many studies have addressed how to improve reasoning. One approach is to provide explicit reasoning examples during inference or training. For example, Wei et al. (2023a) demonstrated that prompting models with structured reasoning instructions, such as chain-of-thought examples or simple cues like *Let's think step by step*, can induce explicit multi-step reasoning traces. Other efforts focus on supervised fine-tuning with reasoning annotations. Early work by Rajani et al. (2019) improved reasoning in smaller models by training on human-written rationales, while later approaches such as STaR (Zelikman et al., 2022) employed self-generated rationales, fine-tuning models on their own successful reasoning trajectories. A complementary line of research addresses the decoding phase, leveraging search algorithms to encourage reasoning exploration. For example, Luo et al. (2024) employed tree-based search, such as Monte Carlo Tree Search, to dynamically explore reasoning paths during generation.

**Reinforcement Learning in LLMs.** Reinforcement learning (RL) has long been central in decision-making tasks (Mnih et al., 2016; 2015; Berner et al., 2019), and is now widely adopted for aligning and fine-tuning LLMs. RL from Human Feedback (RLHF) for LLMs was introduced in InstructGPT (Ouyang et al., 2022b) and subsequently refined by Anthropic (Bai et al., 2022). RLHF has become a cornerstone in training pipelines for models such as Claude 3 (Anthropic, 2024), Gemini (Anil et al., 2023), and GPT-4 (OpenAI, 2023). Typically, RLHF involves supervised fine-tuning, a reward model, and Proximal Policy Optimization (PPO) (Schulman et al., 2017a). PPO stabilizes training by constraining updates through a clipped surrogate objective, providing a practical alternative to Trust Region Policy Optimization (TRPO) (Schulman et al., 2015). Nonetheless, PPO remains sensitive to reward scaling and prone to instability (Wang et al., 2019; Garg et al., 2021; Moalla et al., 2024), which has motivated refinements such as TRGPPO (Wang et al., 2019), alphaPPO (Xu et al., 2023), and PPO-ALR (Jia et al., 2024). Recent analyses further question whether the standard RL challenges motivating PPO apply in the LLM setting: Ahmadian et al. (2024) argue that pretrained LLMs represent strong policies whose variance properties differ substantially from typical RL agents, suggesting that simpler policy-gradient methods may suffice.

**Advancements and Limitations in GRPO.** DeepSeek introduced Group Relative Policy Optimization (GRPO) (Shao et al., 2024b; Guo et al., 2025) to eliminate the critic model by leveraging relative rewards across multiple responses. This technique achieves state-of-the-art performance on math benchmarks and demonstrates that reasoning can emerge as a by-product of reinforcement learning (Guo et al., 2025). Specifically, by training on problems with verifiable answers (e.g., mathematics) using simple correctness and format rewards, GRPO-based models autonomously learned to extend their reasoning length, effectively allocating more compute to reasoning — a phenomenon now referred to as inference-time scaling. As a result, GRPO has become a de facto standard for inducing reasoning in LLMs.

However, emerging studies identify several limitations. Bias effects can skew relative comparisons (He et al., 2025), gradient imbalance may undertrain rare yet informative tokens (Yang et al., 2025; Liu et al., 2025a), and, as in PPO, model performance can degrade or collapse (Dohare et al., 2023). To address these issues, multiple variants have been proposed, including efficiency-focused methods (e.g., CPPO (Lin et al., 2025a)), stability-oriented designs (e.g., S-GRPO (Dai et al., 2025)), and token-level conflict resolution (e.g., GTPO (Simoni et al., 2025)). Complementary analyses deepen our understanding of token-sharing conflicts across completions, as well as policy-collapse mechanisms, highlighting the limits of KL-based constraints and motivating entropy-based approaches (Cui et al., 2025).

**Positioning of this Work.** Building on these developments, we conduct a systematic analysis of GRPO with a focus on mathematical reasoning tasks. We identify which components of the algorithm are essential and which can be simplified without loss of performance. In particular, we investigate whether a REINFORCE-based approach with group-relative advantages — our proposed REINFORCE with Group Relative Advantage (RGRA) — can match or surpass GRPO, offering a more transparent and efficient alternative for reasoning-focused post-training.

## 2.2 PRELIMINARIES

Reinforcement Learning (RL) is a subfield of machine learning in which an agent learns by interacting with its environment in order to improve its performance over time. The goal is to find the policy $\pi$ that maximizes the expected cumulative reward, which can be expressed as:

$$J(\theta) = \mathbb{E}_{\tau \sim \pi_\theta} [G(\tau)]$$

where $G(\tau)$ denotes the return of a trajectory $\tau$ sampled from policy $\pi_\theta$.

**REINFORCE and Policy Gradient Methods** For large language models, the reinforcement learning algorithms most commonly used in the post-training phase belong to the class of policy gradient methods. These methods optimize the policy directly by adjusting its parameters through gradient ascent in the direction that maximizes the expected reward:

$$\theta \leftarrow \theta + \alpha \nabla_\theta J(\theta),$$

where $\alpha$ is the learning rate and $\nabla_\theta J(\theta)$ denotes the policy gradient. The policy gradient can be expressed as:

$$\nabla_\theta J(\theta) = \mathbb{E}_{\tau \sim \pi_\theta} [\nabla_\theta \log \pi_\theta(\tau) G(\tau)]$$

Here, $\pi_\theta(\tau)$ denotes the probability of generating the entire trajectory $\tau = (s_0, a_0, s_1, a_1, \ldots, s_T, a_T)$ under the policy, and $G(\tau)$ represents the return, that is, the cumulative reward associated with the full trajectory. When $G(\tau)$ is estimated through Monte Carlo rollouts of complete episodes, the method corresponds to the REINFORCE algorithm, originally introduced in Williams (1992).

**PPO** A widely used algorithm in the post-training phase of large language models is Proximal Policy Optimization (PPO). PPO is an actor–critic method that stabilizes training and reduces variance in reinforcement learning by employing a clipped surrogate objective. In this setting, the actions $a_t$ correspond to the output tokens $o_t$, while the state $s_t$ is defined by the prompt $q$ together with the previously generated tokens $o_{<t}$.

$$J_{\text{PPO}}(\theta) = \mathbb{E} [q \sim P(Q), \ o \sim \pi_{\theta_{\text{old}}}(O \mid q)]$$

$$\frac{1}{|o|} \sum_{t=1}^{|o|} \left\{ \min \left[ \frac{\pi_\theta(o_t \mid q, o_{i,<t})}{\pi_{\theta_{\text{old}}}(o_t \mid q, o_{i,<t})} A_t, \ \text{clip} \left( \frac{\pi_\theta(o_t \mid q, o_{i,<t})}{\pi_{\theta_{\text{old}}}(o_t \mid q, o_{i,<t})}, \ 1 - \epsilon, \ 1 + \epsilon \right) A_t \right] \right\}$$

Here, $\pi_\theta$ and $\pi_{\theta_{\text{old}}}$ denote the current and previous policy models, respectively. The parameter $\epsilon$ is the clipping coefficient, which constrains the policy ratio to the interval $(1 - \epsilon, 1 + \epsilon)$. The advantage function $A_t$ is typically estimated using Generalized Advantage Estimation (GAE) Schulman et al. (2018), which relies on the rewards $r_t$ for each time step and a value function. The value function, usually parameterized by a model of comparable size to the policy model, acts as a baseline when computing the advantage.

**GRPO** Group Relative Policy Optimization improves RL for LLMs by observing that the value model can be omitted, and the baseline can instead be inferred directly from group statistics. In

particular, the advantage estimation is computed as:

$$\hat{A}_{i,t} = \frac{r_i - \text{mean}(r_1, ..., r_G)}{\text{std}(r_1, ..., r_G)}$$

where $r_i$ denotes the reward assigned to output $o_i$ within the group of $G$ sampled outputs $(o_1, ..., o_G)$ generated from the same prompt. Moreover, the KL penalty with respect to a reference model, originally applied to the reward at each token Shao et al. (2024a), is instead incorporated directly into the loss function. This results in the following loss:

$$J_{\text{GRPO}}(\theta) = \mathbb{E}\left[q \sim P(Q), \{o_i\}_{i=1}^{G} \sim \pi_{\theta_{\text{old}}}(O \mid q)\right] \tag{1}$$

$$\frac{1}{G} \sum_{i=1}^{G} \frac{1}{|o_i|} \sum_{t=1}^{|o_i|} \left\{ \min\left[ r_{i,t}\hat{A}_{i,t}, \ \text{clip}\left(r_{i,t}, \ 1-\epsilon, \ 1+\epsilon\right) \hat{A}_{i,t} \right] - \beta \text{D}_{\text{KL}}\left[\pi_\theta \| \pi_{\text{ref}}\right] \right\}$$

where

$$r_{i,t} = \frac{\pi_\theta(o_{i,t} \mid q, o_{i,<t})}{\pi_{\theta_{\text{old}}}(o_{i,t} \mid q, o_{i,<t})}.$$

and $\beta$ is the parameter controlling the KL regularization.

**RAFT** RAFT aims to provide an alternative to traditional RL methods for LLMs. The proposed algorithm operates by sampling multiple responses for each prompt in a batch, ranking these generated completions using the rewards, and then selecting the highest-ranked response for each prompt. These top responses are then used to construct a new dataset, which serves as the training data for supervised fine-tuning using cross-entropy loss.

## 3 EXPERIMENTS

### 3.1 EXPERIMENTAL SETUP

**Training Datasets** We constructed our training set using problems drawn from GSM8K (Cobbe et al., 2021), a widely used benchmark for grade-school mathematics reasoning. From the training split of the dataset we randomly sampled 1,800 instances. This dataset was selected as it has been explicitly decontaminated from the training corpora of the models employed in our study (Qwen et al., 2025), ensuring unbiased evaluation. It serves as the basis for a quantitative assessment of both performance improvements and training stability.

**Benchmarks** To comprehensively assess the emerging capabilities of the different models in reasoning tasks, we select nine different benchmarks that reflect a diverse level of complexity. We consider five Math-English benchmarks: the testing split of **GSM8K** Cobbe et al. (2021), **MATH** (Hendrycks et al., 2021b), **Gaokao2023-Math-En** Liao et al. (2024), **OlympiadBench** (He et al., 2024), **AMC23** Yang et al. (2024). Then, we consider two Chinese Math benchmarks **CMATH** Wei et al. (2023b) and the **CN-Middle-School** Yang et al. (2024). Finally, we consider two STEM benchmarks: **MMLU-STEM** (English) Hendrycks et al. (2021a) and **Gaokao2024** (Chinese)Zhong et al. (2023).

**Models** We evaluate the proposed training schemes on two instruction-tuned variants of the Qwen2.5 family (Qwen et al., 2025), specifically the 0.5B and 1.5B parameter models and the instruction-tuned 1B paramenters model of the Llama3.2 family Grattafiori et al. (2024). Those models are trained on the GSM8K benchmark, enabling a comparative analysis across small scales.

**Evaluation** To evaluate our model, we calculate the accuracy on a variety of standard benchmark datasets, including English and Chinese. We evaluate the benchmark accuracy of the different models, to understand the capabilities of the models to generalize over different tasks. We also consider training metrics such as average response length, which allows to monitor the ability to learn reasoning, and the average reward obtained by the models during training.

**Experimental Details** To fine-tune the models, we employ LoRA with a rank of 128, effectively reducing the number of trainable parameters to approximately 10% of the original model size. In addition, we incorporate other efficiency techniques such as gradient accumulation and gradient checkpointing. For inference, we adopt VLLM as the underlying engine. For each prompt in the

dataset, we generate a group of 8 completions. The maximum number of tokens generated per completion is set to 512. For the reward system, we implemented two distinct reward signals. The first is a format reward, granting 0.1 points to outputs that follow the specified format. The second is a correctness reward, awarding 1 point for answers that correctly solve the given task. A complete list of experimental parameters can be found in Appendix A.

## 3.2 EXPERIMENTS

We present a series of experiments designed to simplify and decompose the GRPO loss function. Our motivation stems from the observation that, although GRPO has proven effective in improving model performance on mathematical tasks, its structure, which combines group-relative advantage estimation, PPO-style clipping, and KL regularization, introduces a level of complexity that may not be strictly necessary for effective learning.

By systematically isolating and removing individual components, we aim to identify which elements are essential for enabling learning and which can be simplified or omitted without significantly degrading performance.

To this end, we examine three distinct variants of GRPO, evaluating how each simplification affects mathematical performance and the emergence of reasoning abilities in LLMs. Specifically, we analyze:

- **Positive-only Advantages**: We examine the effect of training exclusively on actions that outperform the current baseline, focusing learning on high-reward behaviors and ignoring negative feedback.

$$\mathcal{J}_{\text{GRPO\_pos}}(\theta) = \mathbb{E}\left[q \sim P(Q), \{o_i\}_{i=1}^{G} \sim \pi_{\theta_{\text{old}}}(O \mid q)\right]$$

$$\frac{1}{G}\sum_{i=1}^{G}\frac{1}{|o_i|}\sum_{t=1}^{|o_i|}\left\{\min\left[r_{i,t}\tilde{A}_{i,t},\ \text{clip}\left(r_{i,t},\ 1-\epsilon,\ 1+\epsilon\right)\tilde{A}_{i,t}\right] - \beta \text{D}_{\text{KL}}\left[\pi_\theta \| \pi_{\text{ref}}\right]\right\}$$

  where the modified advantage term is given by:

$$\tilde{A}_{i,t} = \begin{cases} \hat{A}_{i,t} & \text{if } \hat{A}_{i,t} > 0 \\ 0 & \text{otherwise} \end{cases}$$

- **RGRA - Removing PPO-style Constraints**: Inspired by Ahmadian et al. (2024), we investigate the necessity of PPO-style clipping. In this variant, we remove policy ratios and clipping, proposing a REINFORCE variant that preserves GRPO's group-relative advantage estimation. We refer to this simplified approach as RGRA, characterized by the following gradient:

$$\nabla_\theta \mathcal{J}_{\text{RGRA}}(\theta) = \mathbb{E}\left[q \sim P(Q),\ \{o_i\}_{i=1}^{G} \sim \pi_\theta(O \mid q)\right] \tag{2}$$

$$\frac{1}{G}\sum_{i=1}^{G}\frac{1}{|o_i|}\sum_{t=1}^{|o_i|}\left\{\nabla_\theta \log \pi_\theta(o_{i,t} \mid q, o_{i,<t}) \cdot \hat{A}_{i,t}\ -\ \beta \nabla_\theta \text{D}_{\text{KL}}[\pi_\theta \| \pi_{\text{ref}}]\right\}$$

- **REINFORCE with Direct Rewards**: In this variant, we start from RGRA, remove the group-relative advantage estimation, and train directly on the raw reward signal.

Following these investigations, we also evaluated the impact of adopting a simpler rejection sampling strategy that uses a simple cross-entropy, RAFT Dong et al. (2023). The results of the proposed fine-tuning techniques are further compared with those achieved by the fine-tuned version of the models considered.

## 4 RESULTS AND DISCUSSION

Figure 1 reports average reward and response length during training on GSM8K for Qwen2.5 0.5B, 1.5B and Llama3.2 1B models across different objectives. Training with positive-only advantages

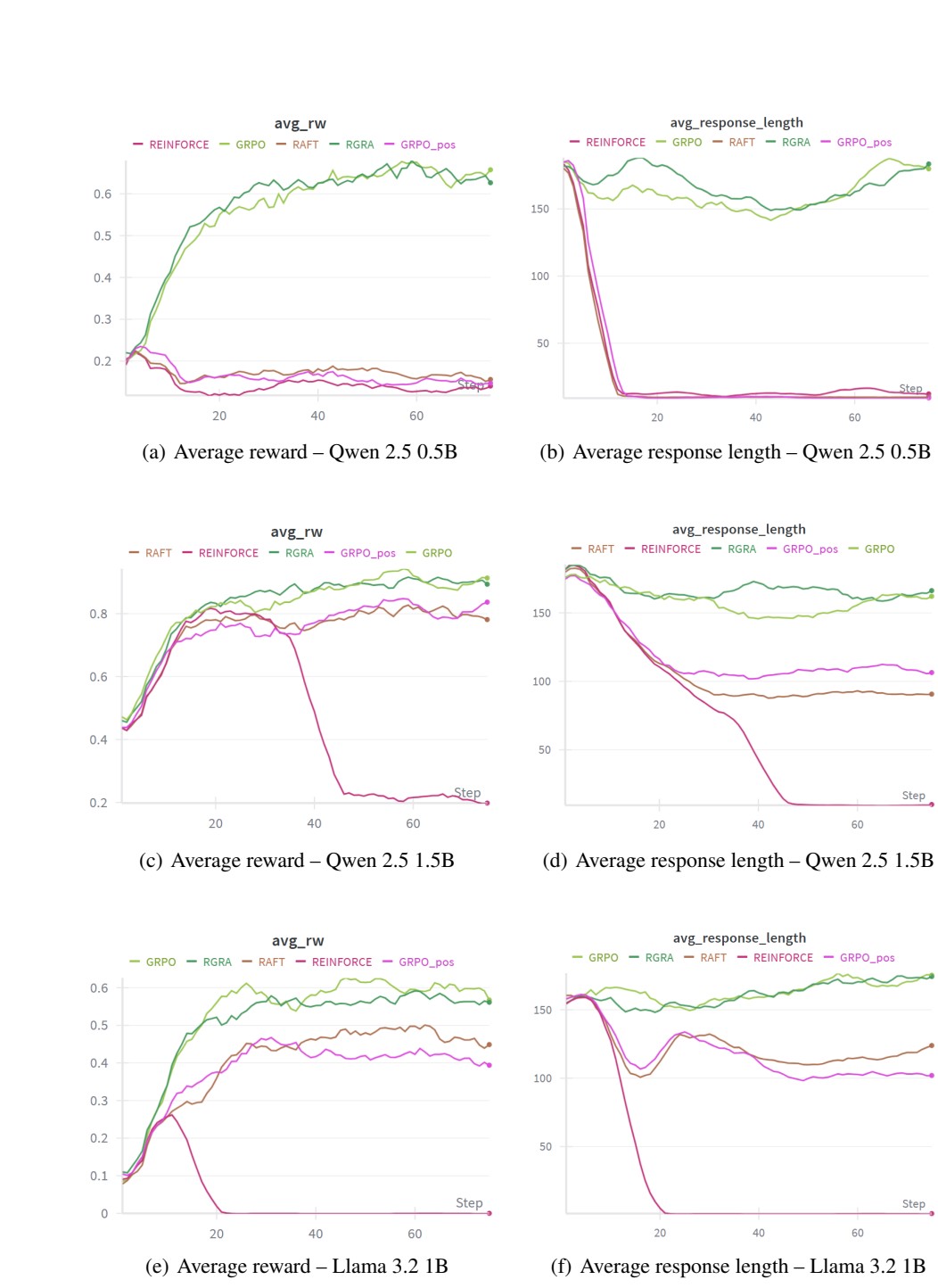

(a) Average reward – Qwen 2.5 0.5B

(b) Average response length – Qwen 2.5 0.5B

(c) Average reward – Qwen 2.5 1.5B

(d) Average response length – Qwen 2.5 1.5B

(e) Average reward – Llama 3.2 1B

(f) Average response length – Llama 3.2 1B

Figure 1: Training metrics (10-step running average).

| Math-English Benchmarks | | | | | | | |
|---|---|---|---|---|---|---|---|
| Model | Method | GSM8K | MATH | Gaokao2023 Math-En | Olympiad Bench | AMC23 | Avg |
| Qwen2.5-0.5-it | - | 41.5 | 22.6 | 21.0 | 6.2 | 7.5 | 19.8 |
| Qwen2.5-1.5-it | - | 61.1 | 38.9 | 35.1 | 11.4 | 17.5 | 32.8 |
| Llama3.2-1.0-it | - | 37.9 | 18.9 | 14.5 | 4.6 | 10.0 | 13.4 |
| Qwen2.5-0.5-it | GRPO | 50.9 | 30.3 | **30.4** | **8.9** | 7.5 | 25.6 |
| Qwen2.5-1.5-it | GRPO | 71.0 | 44.2 | 38.7 | **12.6** | 20.0 | 37.3 |
| Llama3.2-1.0-it | GRPO | 43.0 | **22.9** | 17.4 | 4.6 | 12.5 | 20.1 |
| Qwen2.5-0.5-it | GRPO-pos | 35.6 | 21.1 | 19.7 | 6.1 | 2.5 | 17.0 |
| Qwen2.5-1.5-it | GRPO-pos | 70.6 | 41.0 | 38.7 | 10.7 | 17.5 | 35.7 |
| Llama3.2-1.0-it | GRPO-pos | 41.3 | 21.8 | 18.2 | 5.0 | 12.5 | 19.8 |
| Qwen2.5-0.5-it | RGRA | **53.1** | **32.1** | 29.1 | 8.3 | **10.0** | **26.5** |
| Qwen2.5-1.5-it | RGRA | **72.7** | **46.7** | **42.6** | 12.0 | 17.5 | **38.3** |
| Llama3.2-1.0-it | RGRA | **43.3** | 21.4 | **19.0** | 5.0 | 12.5 | **20.2** |
| Qwen2.5-0.5-it | RAFT | 14.1 | 12.0 | 10.9 | 4.0 | 2.5 | 8.7 |
| Qwen2.5-1.5-it | RAFT | 67.0 | 40.0 | 36.6 | 11.3 | **25.0** | 36.0 |
| Llama3.2-1.0-it | RAFT | 41.8 | 21.0 | 17.4 | 4.7 | 10.0 | 19.0 |
| Qwen2.5-0.5-it | REINFORCE | 44.7 | 26.1 | 24.7 | 4.9 | 12.5 | 22.6 |
| Qwen2.5-1.5-it | REINFORCE | 63.6 | 37.6 | 31.9 | 8.7 | 12.5 | 30.9 |
| Llama3.2-1.0-it | REINFORCE | 41.1 | 22.0 | 17.7 | 4.0 | 10.0 | 19.0 |
| Qwen2.5-0.5-it | ft | 39.5 | 20.7 | 20.8 | 5.0 | 7.5 | 18.7 |
| Qwen2.5-1.5-it | ft | 63.8 | 33.2 | 28.1 | 10.7 | 2.5 | 27.7 |
| Llama3.2-1.0-it | ft | 33.8 | 17.7 | 13.0 | 4.7 | 5.0 | 14.8 |

Table 1: Performance of trained models on English Math benchmarks. All models are trained using gsm8k Dataset.

| Chinese Math Benchmarks | | | | |
|---|---|---|---|---|
| Model | Method | CMATH | CN-Middle-School | Avg |
| Qwen2.5-0.5-it | - | 34.3 | 42.6 | 38.5 |
| Qwen2.5-1.5-it | - | 52.3 | 51.5 | 51.9 |
| Llama3.2-1.0-it | - | 29.5 | 24.8 | 27.2 |
| Qwen2.5-0.5-it | GRPO | 51.2 | 51.5 | 51.4 |
| Qwen2.5-1.5-it | GRPO | **75.0** | 56.4 | 65.7 |
| Llama3.2-1.0-it | GRPO | 33.5 | **26.7** | 30.1 |
| Qwen2.5-0.5-it | GRPO-pos | 46.3 | 36.6 | 41.4 |
| Qwen2.5-1.5-it | GRPO-pos | 71.2 | 59.4 | 65.3 |
| Llama3.2-1.0-it | GRPO-pos | **35.7** | 24.8 | **30.3** |
| Qwen2.5-0.5-it | RGRA | **54.8** | **55.4** | **55.1** |
| Qwen2.5-1.5-it | RGRA | 72.3 | **66.3** | **69.3** |
| Llama3.2-1.0-it | RGRA | 27.5 | 25.7 | 26.6 |
| Qwen2.5-0.5-it | RAFT | 35.2 | 32.7 | 34.0 |
| Qwen2.5-1.5-it | RAFT | 66.2 | 55.4 | 60.8 |
| Llama3.2-1.0-it | RAFT | 34.8 | 21.8 | 28.3 |
| Qwen2.5-0.5-it | REINFORCE | 42.7 | 43.6 | 43.2 |
| Qwen2.5-1.5-it | REINFORCE | 71.0 | 55.4 | 63.2 |
| Llama3.2-1.0-it | REINFORCE | 30.0 | 25.7 | 27.9 |
| Qwen2.5-0.5-it | ft | 29.2 | 42.6 | 35.9 |
| Qwen2.5-1.5-it | ft | 65.3 | 53.5 | 59.4 |
| Llama3.2-1.0-it | ft | 26.2 | 15.8 | 21.0 |

Table 2: Performance of trained models on Chinese Math benchmarks. All models are trained using gsm8k Dataset.

| STEM Benchmarks | | | | |
|---|---|---|---|---|
| Model | Method | English (MMLU-STEM) | Chinese (Gaokao2024) | Avg |
| Qwen2.5-0.5-it | - | 40.6 | 24.4 | 32.5 |
| Qwen2.5-1.5-it | - | 59.2 | 31.5 | 45.4 |
| Llama3.2-1.0-it | - | 31.7 | 13.7 | 22.7 |
| Qwen2.5-0.5-it | GRPO | 41.3 | 21.2 | 31.3 |
| Qwen2.5-1.5-it | GRPO | 58.7 | 32.6 | 45.7 |
| Llama3.2-1.0-it | GRPO | 32.6 | **17.2** | **24.9** |
| Qwen2.5-0.5-it | GRPO-pos | 39.7 | 19.6 | 29.7 |
| Qwen2.5-1.5-it | GRPO-pos | 59.5 | 33.9 | 46.7 |
| Llama3.2-1.0-it | GRPO-pos | 32.4 | 12.8 | 22.6 |
| Qwen2.5-0.5-it | RGRA | **42.0** | **26.5** | **34.3** |
| Qwen2.5-1.5-it | RGRA | **60.1** | **41.2** | **50.7** |
| Llama3.2-1.0-it | RGRA | **33.5** | 11.4 | 22.5 |
| Qwen2.5-0.5-it | RAFT | 39.7 | 20.2 | 30.0 |
| Qwen2.5-1.5-it | RAFT | 58.2 | 33.9 | 46.1 |
| Llama3.2-1.0-it | RAFT | 31.6 | 14.0 | 22.8 |
| Qwen2.5-0.5-it | REINFORCE | 41.1 | 25.7 | 33.4 |
| Qwen2.5-1.5-it | REINFORCE | 57.9 | 31.1 | 44.5 |
| Llama3.2-1.0-it | REINFORCE | 32.8 | 11.1 | 22.0 |
| Qwen2.5-0.5-it | ft | 39.4 | 17.1 | 28.3 |
| Qwen2.5-1.5-it | ft | 55.5 | 24.4 | 40.0 |
| Llama3.2-1.0-it | ft | 31.7 | 12.9 | 22.3 |

Table 3: Performance of trained models on STEM benchmarks (English: MMLU, Chinese: Gaokao2024). All models are trained using gsm8k Dataset.

and RAFT exhibits severe instability, particularly in the 0.5B model, where both reward and response length collapse within the first 20 steps. This collapse manifests as degenerate outputs of minimal length, indicating a reward-hacking phenomenon where the model exploits the absence of negative feedback by converging toward trivial responses. Although the 1.5B and 1B models trained under these regimes avoid immediate collapse, they still demonstrate reward stagnation and gradual shortening of responses, suggesting that discarding negative feedback systematically biases models toward under-exploration and degraded reasoning. By contrast, GRPO and RGRA maintain stable training dynamics in both model sizes, both achieving comparable reward trajectories. These findings reinforce that advantage estimation is essential for stabilizing reinforcement learning in LLMs, while PPO-style clipping is not strictly required when initializing from strong policies. However, training with direct REINFORCE on raw rewards collapses even in the larger 1.5B model, underscoring the indispensable role of advantage estimation for stability.

**Math-English Benchmarks** The performance of trained models on English mathematical reasoning tasks is reported in Table 1. First, we can observe that models trained with RAFT or positive-only GRPO show worse performance compared to both GRPO and RGRA. Especially for the Qwen2.5 0.5B model. This poor result aligns with the observed training collapse and response truncation. Second, GRPO achieves consistent improvements over instruction-tuned baselines, particularly on GSM8K and MATH, but RGRA outperforms GRPO in most settings. Specifically, RGRA achieves the highest average performance across the Math-English benchmarks for the three models used, surpassing GRPO in 17 out of 27 individual comparisons. This further supports the claim that PPO-style constraints are not necessary for effective learning from strong initialization.

**Chinese Math Benchmarks** The results on CMATH and CN-Middle-School (Table 2) reveal a similar pattern. RAFT and positive-only GRPO exhibit weaker performance with respect to GRPO. In contrast, RGRA delivers the strongest results in the Qwen2.5 models, achieving average accuracies of 55.1 (0.5B) and 65.3 (1.5B), exceeding both standard GRPO and fine-tuned baselines.

**STEM Benchmarks** Performance in STEM-related evaluations (MMLU-STEM and Gaokao2024-STEM, see Table 3) further illustrates the limitations of training regimes that ignore negative feedback. RAFT and positive-only GRPO again fail to surpass fine-tuned baselines. GRPO

shows modest improvements and outperforms the other methods on Llama3.2. By contrast, RGRA achieves the best improvements on the Qwen2.5 models.

**Emergence of Reasoning Behaviors** Beyond raw benchmark scores, training methods also differ in their ability to induce reasoning behaviors. On the Countdown dataset, we observe that RAFT and positive-only GRPO models fail to generate explicit reasoning steps, instead outputting direct final answers (Figure 2(a)). Conversely, GRPO and RGRA models exhibit emergent reasoning, including explicit re-evaluation of intermediate answers (Figure 2(b)). This highlights that robust training regimes not only stabilize learning and improve benchmark scores but also foster the development of interpretable reasoning strategies in LLMs.

<answer> **35 - 19 + 44** </answer>

Firstly, we need to find a way to get close to 64 using the given numbers: 63, 95, and 96. We notice that if we subtract 95 from 63, we get -32. However, since we cannot directly use negative numbers or fractions with these specific numbers, **let's try another approach**. If we add 96 to 95, we get 191, which is too large. **Instead, let's consider** adding 96 to 63 but then subtracting 95: <answer>(63 + 96) - 95</answer> This gives us: (63 + 96) - 95 = 159 - 95 = 64 So, the final equation is: <answer>**(63 + 96) - 95**</answer>

(a) Answer without reasoning trace.

(b) Answer with reasoning trace.

Figure 2: Comparison of answers with and without reasoning traces.

## 5 CONCLUSION

In this work, we investigated the GRPO loss function with the goal of disentangling its components and identifying which are essential for effective post-training of large language models. Through a series of controlled ablations and benchmark evaluations, we analyzed the impact of retaining or removing specific elements of the objective on learning stability, mathematical reasoning, and generalization across tasks and languages.

Our results demonstrate three key findings. First, negative feedback is indispensable: methods that ignore it—such as RAFT or positive-only GRPO—exhibit instability, collapse, and consistently degraded performance. Second, advantage estimation is crucial: removing it, as in REINFORCE with direct rewards, destabilizes learning even in larger models with strong initial policies. Third, PPO-style clipping is unnecessary: eliminating clipping and policy ratios does not harm stability; instead, it simplifies training and can lead to improved performance.

These insights motivated the introduction of RGRA, a simplified variant of GRPO that discards PPO-style constraints while preserving group-relative advantage estimation. Across training dynamics, multilingual mathematical benchmarks, STEM evaluations, and reasoning behavior analyses, RGRA not only achieves stable learning but also surpasses GRPO on 17 over 27 tasks, establishing it as a competitive reinforcement learning objective for reasoning tasks.

Overall, this study advances our understanding of how reinforcement learning objectives shape the post-training of large language models. By showing that GRPO can be simplified without sacrificing performance, we lay the groundwork for future research on reinforcement learning strategies that further enhance reasoning capabilities and generalization in LLMs.

Future works will consider exploring additional tasks outside the mathematical domain. An additional research work could address larger models, which was not possible here due to hardware constraints.

## 6 REPRODUCIBILITY STATEMENT

We provide detailed descriptions of each algorithm in Section 3 and Appendix A, including the techniques, fine-tuned hyperparameters, and infrastructures used in our experiments. The link to our code is https://anonymous.4open.science/r/math_llms-FE4E/README.md.

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

## A HYPERPARAMETERS USED

Table 4 summarizes the training hyperparameters.

Table 4: Hyper-parameters used for training on GSM8K.

| Experiment | Hyper-parameter | Value |
|---|---|---|
| GRPO | Batch size | 24 |
| | Learning rate | $1 \times 10^{-5}$ |
| | Group size $G$ | 8 |
| | Temperature | 1 |
| | KL coefficient | 0.005 |
| | Max new tokens | 512 |
| | Warmup steps | 5 |
| GRPO_pos | Batch size | 24 |
| | Learning rate | $1 \times 10^{-5}$ |
| | Group size $G$ | 8 |
| | Temperature | 1 |
| | KL coefficient | 0.005 |
| | Max new tokens | 512 |
| | Warmup steps | 5 |
| RGRA | Batch size | 24 |
| | Learning rate | $1 \times 10^{-5}$ |
| | Group size $G$ | 8 |
| | Temperature | 1 |
| | KL coefficient | 0.005 |
| | Max new tokens | 512 |
| | Warmup steps | 5 |
| REINFORCE | Batch size | 24 |
| | Learning rate | $1 \times 10^{-5}$ |
| | Group size $G$ | 8 |
| | Temperature | 1 |
| | KL coefficient | 0.005 |
| | Max new tokens | 512 |
| | Warmup steps | 5 |
| RAFT | Batch size | 24 |
| | Update epochs per stage | 1 |
| | Learning rate | $1 \times 10^{-5}$ |
| | Group size $G$ | 8 |
| | Temperature | 1 |
| | Max new tokens | 512 |
| | Warmup steps | 5 |
| SFT | Learning rate | $1 \times 10^{-4}$ |
| | Decay mode | Linear |
| | Epochs | 1 |
| | Batch size | 8 |
| | Warmup steps | 5 |

