# OpenReview forum: "Are complicated loss functions necessary for teaching LLMs to reason?"
_ICLR.cc/2026/Conference — Submitted to ICLR 2026_

### Official Review · Reviewer_7cYU · 2025-10-29

**Soundness:** 3
**Presentation:** 3
**Contribution:** 3
**Rating:** 6
**Confidence:** 4

**Summary:**

This paper investigates the necessity of complex loss functions, specifically Group Relative Policy Optimization (GRPO), for enhancing the reasoning capabilities of Large Language Models (LLMs). The authors conduct a systematic analysis of GRPO, an algorithm that combines group-relative advantage estimation, PPO-style clipping, and KL regularization.
The paper identifies two key findings:
Negative feedback is essential. Training solely on actions that outperform a baseline (i.e., positive-only advantages) or using simpler rejection sampling (RAFT) leads to training instability, performance collapse, and a failure to elicit reasoning behaviors.
PPO-style constraints are unnecessary. The analysis demonstrates that PPO-style components, such as policy ratio clipping, are not required to improve mathematical reasoning performance or maintain training stability.
Experiments across standard mathematical benchmarks indicate that RGRA achieves stable training dynamics and demonstrates stronger performance than the more complex GRPO, surpassing it in 17 out of 27 task comparisons.

**Strengths:**

This paper's originality is strong. The authors challenge the assumed necessity of all components within the successful GRPO framework . This "less is more" approach, which rigorously questions the utility of established components like PPO-style clipping, represents an original and valuable methodological contribution. The claims are substantiated by a comprehensive and robust body of empirical evidence, including extensive quantitative benchmarks across multiple model families and languages (Tables 1-3) , crucial analysis of training dynamics and stability (Figure 1) , and insightful qualitative analysis of emergent reasoning behaviors (Figure 2). The experimental setup is sound and provides convincing validation for the paper's conclusions.

**Weaknesses:**

The paper's primary weakness lies in its significant overgeneralization of claims from a narrow and limited experimental setup. The central conclusion that PPO-style constraints are "unnecessary" for teaching reasoning is drawn exclusively from experiments on small-scale models, ranging from 0.5B to 1.5B parameters. PPO's clipping mechanism was precisely designed to ensure stability during large, high-variance policy updates, which are a far greater concern in the state-of-the-art 70B+ models. The paper provides no evidence that its findings would hold in a large-scale setting, thus failing to adequately support its ambitious and broad claims.
Furthermore, the dismissal of baseline methods like RAFT and positive-only GRPO as inherently unstable is unconvincing. Their catastrophic collapse (shown in Figure 1) is observed on a minuscule training dataset of only 1,800 instances , which is highly susceptible to reward-hacking. More importantly, the paper fails to provide evidence of rigorous hyperparameter tuning for these baselines. Their collapse could simply be an artifact of a poorly chosen learning rate or insufficient KL regularization, rather than a fundamental flaw in the methods themselves. Without a proper hyperparameter sweep to find the most stable configuration for these baselines, the paper's conclusion is not fully substantiated.

**Questions:**

The paper's central claim that PPO-style constraints are "unnecessary" is derived from experiments on relatively small-scale models (0.5B to 1.5B). Given that PPO's clipping mechanism was specifically designed to ensure stability for large scale policies with high variance updates, what justification or evidence can you provide that this finding will generalize to the 70B+ or 100B+ models where such stability constraints are traditionally considered critical?
The experiments are confined entirely to mathematical reasoning, a domain characterized by sparse and verifiable binary reward signals (i.e., correct/incorrect). How do you anticipate the stability of RGRA (which lacks clipping) will hold in standard alignment scenarios (e.g., helpfulness, safety) that rely on dense, non stationary, and often noisy rewards from learned preference models?
The paper concludes that methods ignoring negative feedback (RAFT, GRPO-pos) are fundamentally unstable, citing their rapid collapse on a small 1,800-instance training set. Could you elaborate on the extent of the hyperparameter search (e.g., learning rate) conducted for these baselines? How can you be certain this collapse is an inherent flaw of the methods, rather than an artifact of sub-optimal tuning or a simple case of reward-hacking on a dataset small enough to be easily exploited?
The results suggest RGRA does not just match, but often outperforms GRPO. Since the key difference is the removal of the PPO clipping and policy ratio terms, what is the mechanism for this performance improvement?
The paper shows that standard REINFORCE with direct rewards collapses, even on the 1.5B model, which you use to underscore the necessity of advantage estimation. Could you clarify the tuning process for this specific baseline? Does this result definitively prove that any REINFORCE-style method without advantage estimation is doomed to fail in this setting, or could this collapse also be sensitive to hyperparameter choices?

---

### Official Review · Reviewer_ZvS5 · 2025-11-01

**Soundness:** 1
**Presentation:** 1
**Contribution:** 1
**Rating:** 0
**Confidence:** 5

**Summary:**

The paper looks into GRPO training for reasoning models. I tests three variants of GRPO 1. positive only reward 2. GRPO without importance sampling 3. naive reinforce

**Strengths:**

N/A

**Weaknesses:**

* There is a severe lack of novelty in the paper. The proposed RGRA is essentially GRPO without importance sampling.
* The paper is poorly composed; the results are not well organized. Figure 1 occupies an entire page without any accompanying analysis in the caption. Tables 1 and 2 are also poorly formatted, lacking proper bolding and explanations for abbreviations. From this standpoint alone, the paper feels far from complete.
* There is almost no discussion regarding the differences between RGRA and GRPO. Why does removing importance sampling and not using a clipping objective lead to better training?
* There is no comparison with related works at all (e.g., Dr.GRPO, DAPO, etc.).

**Questions:**

* Overall, the paper lacks a clear motivation, shows limited novelty, and provides insufficient analysis of the results. Major revisions are needed.

---

### Official Review · Reviewer_nvCs · 2025-11-01

**Soundness:** 2
**Presentation:** 2
**Contribution:** 2
**Rating:** 0
**Confidence:** 3

**Summary:**

This paper does an ablation over the components of the GRPO loss function, namely advantage clipping, negative examples, and KL regularization.

**Strengths:**

This paper studies an important question in RL post-training, namely which components are required in the loss function to get the models to perform well. Based on their findings, the authors propose RGRA for LLM post training.

**Weaknesses:**

There are several major weaknesses with this paper. To begin, the framing of the paper is an ablation over the main components of the GRPO loss. However, there are several key components missing from this ablation:
- as far as I understand, the authors do not sweep over the hyperparameters of any of the baselines they run. Critically, for an ablation over components of GRPO, they do not sweep over the number of rollouts, nor over the amount of steps taken off policy by the algorithm (I am referring to doing multiple gradient steps over a set of rollouts for a given batch)
- the authors start from pretrained models, which can confound the results. Namely, [1] shows that the KL regularizer can have different effects based on the pretraining data
- the baselines are quite trivial, especially with respect to the length of the chain of thought required to solve the prompts. In general, the terms in the loss start to be significant for long chains, or the number of offline steps taken by the algorithm

Based on the above comments, I believe proposing RGRA is not currently backed by empirical results.


[1] Echo Chamber: RL Post-training Amplifies Behaviors Learned in Pretraining

**Questions:**

No questions.

---

### Official Review · Reviewer_TxYV · 2025-11-01

**Soundness:** 2
**Presentation:** 3
**Contribution:** 1
**Rating:** 2
**Confidence:** 4

**Summary:**

This paper studies different components of the GRPO (Group Relative Policy Optimization) loss in improving reasoning in LLMs. The authors break down the loss into its main components—negative feedback, PPO-style clipping, and advantage estimation—and test simplified versions such as positive-only GRPO, their proposed REINFORCE with Group Relative Advantage (RGRA), and direct REINFORCE. Their experiments on small models (Qwen2.5-0.5B/1.5B and Llama3.2-1B) show that removing negative feedback leads to collapse and reward stagnation and PPO clipping can be dropped without hurting performance.

**Strengths:**

* Systematic studies like the one that the paper conducts is generally important for the community, especially for understanding RL post-training for LLMs.
* Authors test on two different model families and take care in evaluating on a comprehensive set of benchmarks split across Chinese/English and math/other subject domains.

**Weaknesses:**

* The model scale and setting (<=1.5B parameter models with LoRA fine-tuning) is limited and it's unclear if their findings extrapolate to larger model scales and full fine-tuning.
* In particular, prior work [1] seems to show a different result that positive-only reinforcement can be competitive with GRPO/PPO provided verifiable rewards are used and poor prompts are filtered. The findings from Xiong et al. are from larger models (7B-70B), which supports the potential limitations of the model size and setting studied in this work.
* The reported results lack confidence intervals and seem necessary to draw strong conclusions like the ones made in this work (in particular, how significant is the performance delta between RGRA and GRPO)? I'm sympathetic to the author's limited compute constraining them to their training setup, but multiple seeds and further performance analysis (eg. pass/majority@k performance) would strengthen their results.

While the paper’s goal of simplifying GRPO is well-motivated, the evidence feels too narrow and limited to support its strong claims about the necessity of negative feedback.

Minor: Some areas in the manuscript need `\citep` (Line 169, 263 to name a few).

[1] Xiong, Wei, et al. "A minimalist approach to llm reasoning: from rejection sampling to reinforce." arXiv preprint arXiv:2504.11343 (2025).

**Questions:**

* The training dataset appears quite small (around 1.8k examples). Could the authors clarify why this size was chosen, and whether you observed any sensitivity to data scale?
* In the positive-only advantage setup, do the authors ensure that each batch contains enough positively rewarded samples for stable gradient estimation?
* In Figure 1(a), both REINFORCE and RAFT collapse only for the Qwen 0.5B model. Do you have an explanation for why this smaller model is unstable compared to the 1.5B and 1B variants? Could this have been mitigated with a cold start stage?
* Regarding clipping, what $\epsilon$ value was used for GRPO runs, and roughly what fraction of updates were actually clipped during training?
* The author's results seem to differ from Xiong et al., who find that positive-only RAFT remains competitive with GRPO. Could you comment on the key differences in setup (e.g., model scale, filtering, or reward structure) or clarify the discrepancy?

---

### Meta-Review · Area_Chair_4A8F · 2025-12-14

**Summary:**

This paper does an ablation study of components of the GRPO loss in improving reasoning in LLMs. The authors have done experiments on small (<=1.5B) LLMs from two families and concluded that negative feedback is necessary and PPO clipping can be dropped without hurting performance.

The reviewers share a few significant concerns on, e.g., limited model scales, missing baselines, and missing discussion w.r.t. recent work which has conflicting conclusions.
Moreover, no rebuttal is provided by the author.

Therefore, I do not think the paper is ready for publication.

**Reviewer Concerns:**

No rebuttal is provided by the authors. Quite a few valid concerns raised by the reviewers are still outstanding.
- The scale of models is too limited to obtain reliable conclusions that could generalize/extrapolate. [Reviewer TxYV and 7cYU]
- Missing competitive baselines [Reviewer nvCs, ZvS5, and 7cYU]
- Missing discussion with recent work [Reviewer TxYV and nvCs]
- Multiple runs are necessary to ensure the results are statistically reliable. [Reviewer TxYV and nvCs]

**Reviewer Scores:**

Since there is no rebuttal provided by the authors, the scores are unchanged.

---

### Decision · Program_Chairs · 2026-01-26

Reject